# Bridging Discovery and Treatment: Cancer Biomarker

**DOI:** 10.3390/cancers17223720

**Published:** 2025-11-20

**Authors:** Hengrui Liu, Ilayda Karsidag, Rebecca Golin, Guangzhen Wu

**Affiliations:** 1Department of Biochemistry, University of Cambridge, Cambridge CB2 1QW, UK; 2School of Biological Sciences, University of California San Diego, San Diego, CA 92093, USA; 3Leon M Goldstein School, New York, NY 11235, USA; 4Department of Urology, The First Affiliated Hospital of Dalian Medical University, Dalian 116011, China

**Keywords:** translational research, cancer biomarkers, tumor heterogeneity, multi-omics technologies, precision oncology, personalized medicine

## Abstract

Cancer is a complicated disease, and every patient’s tumor is different. Biomarkers are small molecular clues, found in blood, tissue, or medical scans, that help doctors detect cancer earlier and choose the most suitable treatments. In this article, we explain the main types of cancer biomarkers and how they are used to guide diagnosis, monitor treatment response, and identify early signs of relapse. We also highlight new technologies, such as liquid biopsies and artificial intelligence, that are making cancer care more precise and less invasive. By bringing these advances together, biomarker-based approaches can improve treatment decisions and ultimately help patients achieve better outcomes.

## 1. Introduction

With the rapid advancement of genomic sequencing, multi-omics technologies, and big-data analytics, the conventional “one-size-fits-all” treatment paradigm based solely on tumor type and stage is increasingly inadequate for addressing the heterogeneous nature of cancer [1]. Precision oncology has emerged to tailor therapies to each patient’s unique molecular profile, tumor microenvironment, and inherited susceptibilities [2]. For example, identifying activating epidermal growth factor receptor (EGFR) mutations in non-small cell lung cancer patients enables clinicians to select EGFR inhibitors for those most likely to respond, thereby improving outcomes while minimizing unnecessary toxicity [3,4].

Biomarkers serve as the critical link between molecular discovery and clinical decision-making, encompassing genetic alterations, epigenetic modifications, protein expression patterns, metabolic signatures, and immune microenvironment indicators [5]. Across all stages of cancer care, from initial diagnosis and risk stratification to treatment monitoring and resistance assessment, biomarkers guide more informed interventions [6,7]. For instance, tracking circulating tumor DNA (ctDNA) levels throughout targeted therapy can reveal emerging resistant clones, allowing for real-time adjustment of treatment strategies [8]. Similarly, measuring programmed death-ligand 1 (PD-L1) expression on tumor cells helps predict which patients are most likely to benefit from immune checkpoint inhibitors [9].

This review aims to provide a comprehensive overview of current cancer biomarker classes and detection platforms, illustrate their diverse clinical applications, and address the translational challenges that arise when moving from laboratory discovery to routine practice. The manuscript is organized as follows: We begin by categorizing biomarkers according to sample type and molecular modality, then examine their clinical applications in disease monitoring, theranostics, and early detection. Next, we explore how biomarkers guide therapeutic decision-making and illuminate mechanisms of treatment resistance. We then address the key translational hurdles, biomarker validation, regulatory approval, and standardization, that must be overcome to bring discoveries into routine practice. In the final section, we consider personalized treatment strategies in light of tumor heterogeneity, assess cost-effectiveness, and outline promising directions for future research..

## 2. Current Cancer Biomarkers and Detection Modalities

### 2.1. Sample Type

#### 2.1.1. Tissue-Based Biomarkers

Tissue biopsies remain the gold standard for definitive cancer diagnosis and for guiding initial therapeutic decisions. Histopathological examination coupled with immunohistochemistry (IHC) reveals the expression of key protein markers, such as HER2 in breast cancer or PD-L1 in non-small cell lung cancer, that directly inform the choice of targeted antibodies or immunotherapies [10]. Meanwhile, genomic profiling of tumor tissue via next-generation sequencing (NGS) panels identifies driver mutations (e.g., EGFR [11], BRAF [12]) and gene fusions (e.g., ALK [13], ROS1 [13]), enabling selection of small-molecule inhibitors that specifically block oncogenic signaling [14,15]. Beyond first-line treatment, repeat biopsies at progression can uncover secondary resistance mutations, such as EGFR T790M [16] or MET amplification [17], that prompts a switch to next-generation inhibitors or combination regimens. In this way, tissue-based biomarkers not only initiate but also continually refine precision therapy (Figure 1).

#### 2.1.2. Liquid Biopsy

Liquid biopsy approaches allow for minimally invasive sampling of circulating tumor DNA (ctDNA) [18], circulating tumor cells (CTCs) [19], and tumor-derived proteins [20] or metabolites [21]. By tracking ctDNA variant allele frequencies over time, clinicians can monitor tumor burden and detect emerging resistance clones weeks to months before changes become apparent on imaging [22]. This real-time molecular surveillance supports adaptive treatment strategies, for example, escalating or switching targeted agents upon an early rise in ctDNA rather than waiting for radiographic progression. Similarly, quantification of CTCs or exosomal RNA markers can predict response to therapy and identify patients who might benefit from enrollment in clinical trials of novel agents. Urine-based assays for specific mutations (such as telomerase reverse transcriptase, TERT, promoter mutations in bladder cancer [23]) further expand the reach of liquid biopsy, facilitating surveillance even when blood draws are impractical (Figure 1).

#### 2.1.3. Imaging Biomarkers

Functional and molecular imaging modalities provide non-invasive, whole-body assessments of tumor biology that complement molecular assays. ^18^F-FDG positron emission tomography–computed tomography (PET/CT) measures glucose uptake to evaluate metabolic activity, with early decreases in standardized uptake value (SUV) serving as a surrogate for therapeutic efficacy, even after a single cycle of chemotherapy or targeted therapy [24]. Newer tracers targeting prostate-specific membrane antigen (PSMA), somatostatin receptors, or fibroblast activation protein enable both visualization and delivery of radiotherapeutic payloads in a theranostic paradigm. Dynamic contrast-enhanced magnetic resonance imaging (MRI) and perfusion CT assess tumor vascularity and permeability, informing the use of anti-angiogenic agents [25,26,27]. By integrating quantitative imaging biomarkers into response criteria, oncologists can optimize treatment duration, avoid futile therapies, and tailor combinations based on each tumor’s unique physiology (Figure 1).

#### 2.1.4. Other Clinical Information

Traditional clinical parameters, such as tumor staging, performance status, and routine laboratory values, remain essential biomarkers that influence treatment planning and risk stratification. For instance, a high neutrophil-to-lymphocyte ratio [28] or elevated lactate dehydrogenase [29] may signal systemic inflammation and poorer prognosis, leading clinicians to pursue more aggressive combination therapies upfront. Microbiological cultures in patients with febrile neutropenia guide the safe continuation of cytotoxic regimens through targeted antimicrobial prophylaxis [30,31]. Moreover, emerging evidence suggests that comorbid conditions and host immune competence, captured through simple blood-count biomarkers [32] and organ-function tests [33], can predict tolerability and guide dose adjustments. In this way, “non-molecular” clinical biomarkers ensure that targeted and immunotherapeutic advances are delivered safely and effectively to the right patients.

### 2.2. Detection “Dimensions” of Biomarkers

Biomarker detection can be classified according to the scale and context in which molecular signals are captured, ranging from bulk assays that average signals across heterogeneous cell populations to highly resolved, spatially aware, and global approaches. Below, we outline five complementary dimensions of detection, including bulk, single-cell, spatial, and global profiling (such as cfDNA), highlighting their distinct advantages, limitations, and example applications.

#### 2.2.1. Local Bulk Profiling Versus Global Bulk Profiling

Local bulk assays, such as bulk RNA-seq, whole-cell or plasma proteomics, and targeted methylation panels, measure the aggregate molecular signal from a discrete tissue biopsy sample. While these methods offer high sensitivity for abundant analytes and relatively low per-sample cost, their reliance on precise sampling introduces significant bias: a needle biopsy may miss microscopic niches of resistant clones or immunologically active regions, and the spatial heterogeneity of solid tumors means that any single sample may not represent the lesion as a whole. In practice, sampling bias can lead to false negatives or underestimation of key biomarkers when analytes are unevenly distributed across the tumor mass [34]. Moreover, by averaging signals over millions of cells, local bulk profiling obscures the molecular signatures of rare subpopulations that often drive disease progression and treatment resistance.

Circulating tumor cells (CTCs) and circulating cell-free DNA (cfDNA) represent global bulk detection approaches that capture tumor-derived material shed into the bloodstream from all anatomical sites. By interrogating not only somatic mutations but also cfDNA fragment length distributions, genome-wide methylation landscapes, and nucleosome positioning patterns, these assays furnish a non-invasive, whole-body portrait of tumor burden capable of detecting minimal residual disease and enabling multi-cancer early detection. For example, pan-cancer screening tests combine targeted mutation panels with epigenomic signatures to flag asymptomatic malignancies, while fragmentomic profiling uncovers subtle shifts in cfDNA topology indicative of residual or recurrent disease following therapy [35]. Although this global sampling is invaluable when the tumor location is unknown, it inherently dilutes tumor-specific signals, reducing sensitivity and elevating background noise, and, because cfDNA originates from diverse tissues (including hematopoietic and inflamed cells), it lacks spatial resolution, often requiring complementary assays or computational deconvolution to accurately localize the source lesion (Figure 2).

#### 2.2.2. Single-Cell Profiling Versus Spatial Profiling

Single-cell technologies, such as single-cell RNA sequencing (scRNA-seq), mass-cytometry (CyTOF), single-cell assay for transposase-accessible chromatin using sequencing (ATAC-seq), and emerging proteomic workflows, resolve the molecular state of individual cells, uncovering rare clones, transitional cell states, and lineage relationships that are invisible in bulk measurements. This cellular granularity has revealed, for example, minor subpopulations with epithelial–mesenchymal transition programs linked to metastasis and chemoresistance, guiding the design of combination therapies to suppress both dominant and resistant niches. Yet single-cell profiling inherits the spatial sampling bias of tissue biopsies, and high per-cell costs limit throughput to thousands rather than millions of cells, potentially overlooking low-frequency populations. Moreover, technical noise and dropout, where low-abundance transcripts or proteins fall below detection thresholds, can distort the true diversity and functional landscape of the microenvironment [34].

Spatially resolved methods integrate molecular profiling with histological context, preserving the native tissue architecture as they measure expression or protein abundance in situ. Platforms such as multiplexed immunofluorescence (e.g., CODEX, CyCIF), spatial transcriptomics (e.g., Visium, GeoMx), and imaging mass cytometry map the organization of tumor cells, stroma, and immune infiltrates at sub-millimeter resolution [36]. By identifying “immune-cold” zones or perivascular niches that foster therapy resistance, spatial profiling informs localized interventions, such as focused radiotherapy or intratumoral immunomodulator delivery, targeted to the regions of greatest need [37]. Nonetheless, these assays incur higher costs and lower sample throughput than bulk approaches, and resolution limits remain: techniques like laser-capture microdissection still perform bulk analysis on small regions and may miss infiltrating cells below their capture threshold, while low-signal markers risk dropout and background noise akin to single-cell workflows (Figure 3).

In parallel, organoid models provide a complementary platform by preserving patient-specific architecture and functional phenotypes ex vivo. Although they cannot fully recapitulate in situ spatial immune or stromal interactions, organoids offer a controllable system for perturbation studies that nicely complements the in-tissue insights gained from spatial profiling.

### 2.3. Detection Molecular Modality

#### 2.3.1. DNA-Level Biomarkers

Point mutations and small insertions/deletions in oncogenes or tumor suppressors are among the most actionable DNA biomarkers. For example, EGFR exon 19 deletions [38] or L858R substitutions [39] in non-small cell lung cancer, predict robust responses to first- and second-generation EGFR tyrosine kinase inhibitors [40]. Likewise, BRAF V600E mutations in melanoma or colorectal cancer identify patients eligible for BRAF/MEK inhibitor combinations [41,42]. Monitoring the allele frequency of these variants in circulating tumor DNA (ctDNA) during therapy enables early detection of resistance-associated mutations (e.g., EGFR T790M), prompting timely switching to next-generation inhibitors [43,44], such as the well-recognized false-negative ctDNA results in low-shedding tumors (e.g., early-stage prostate or pancreatic cancers), which can obscure emerging resistance [45].

On the other hand, gene amplifications and deletions can drive tumor growth or therapy resistance. HER2 amplification in breast [46] and gastric [47] cancers are the basis for anti-HER2 antibodies and antibody–drug conjugates [48]. Conversely, MET amplification emerging after EGFR inhibitor therapy serves as a mechanism of acquired resistance [17]. Recognizing this CNV shift allows clinicians to add MET inhibitors or adjust treatment regimens. High-level MYC or CCND1 amplifications may also forecast sensitivity to CDK4/6 inhibitors [49] or BET bromodomain inhibitors [50] under investigation.

In liquid biopsy applications, fragmentomics has rapidly emerged as a widely adopted technique [51]. Analysis of DNA fragment size patterns in plasma enhances tumor specificity beyond simple mutation detection. Tumor-derived ctDNA fragments often display characteristic size profiles (e.g., shorter fragment lengths [52], altered end motifs [53]) compared with normal cell-free DNA. By enriching for these tumor-specific fragments, fragmentomic assays achieve more sensitive minimal residual disease (MRD) detection after surgery or adjuvant therapy. Detecting residual ctDNA months before clinical relapse enables preemptive intensification of systemic therapy to prevent overt recurrence.

In addition, extracellular vesicles (EVs) such as exosomes protect high-molecular-weight genomic DNA shed from tumor cells [54]. Unlike highly fragmented ctDNA, EV-DNA can span entire introns and exons, preserving structural variants and gene fusions that may be missed by standard cfDNA assays. For instance, capturing ALK [55] or ROS1 [56] fusion breakpoints in EV-DNA from lung cancer patients allow identification of candidates for ALK/ROS1 inhibitors even when tissue biopsy is infeasible. As therapies targeting these rearrangements achieve durable remissions, EV-DNA profiling represents a powerful tool to both discover and monitor actionable structural alterations (Figure 4).

#### 2.3.2. RNA-Level Biomarkers

Profiling mRNA expression in tumor tissue or circulating cell-free RNA offers a direct snapshot of the oncogenic programs active within cancer cells, and this information can be immediately translated into treatment selection. For example, high transcript levels of VEGFA in a patient’s tumor may indicate a dependency on angiogenic signaling [57]. Such patients are prime candidates for agents that target the VEGF pathway, allowing clinicians to tailor anti-angiogenic therapy rather than applying it broadly. Similarly, quantifying PD-L1 mRNA in biopsy specimens can uncover patients who might benefit most from checkpoint blockade, even when protein IHC results are equivocal [58,59]. Beyond single markers, multi-gene expression signatures that include proliferation and survival genes, such as MKI67 [60,61], PCNA [62], and BCL2 [63] family members can be used to monitor molecular response: a precipitous drop in these transcripts shortly after therapy initiation often presages tumor regression on imaging, enabling early dose de-escalation in responders or swift regimen changes in non-responders to avoid ineffective treatment, such as instability of circulating RNA transcripts and dropout of low-abundance mRNAs that can lead to underestimation of key pathways [64].

Long noncoding RNAs (lncRNAs) orchestrate complex regulatory networks that underpin tumor progression and therapy resistance [65]. Aberrant overexpression of lncRNAs such as HOTAIR [66,67] and MALAT1 [68,69,70] has been implicated in metastatic spread and poor response to chemotherapy across multiple tumor types. By measuring lncRNA levels in tumor biopsies, oncologists can identify patients at high risk for early relapse who may derive greater benefit from intensified adjuvant regimens or inclusion in trials of novel agents. Moreover, the discovery of lncRNAs that drive resistance pathways has given rise to targeted antisense therapies aimed at silencing these transcripts; early-phase studies of HOTAIR-directed oligonucleotides, for example, seek to restore chemosensitivity in refractory cancers by dismantling the epigenetic scaffolds that these lncRNAs establish [71].

Extracellular vesicles (EVs) serve as vessels conveying a rich cargo of RNA species, including mRNA fragments, miRNAs, and lncRNAs, from tumors into the circulation, providing a minimally invasive biopsy [72,73]. EV-encapsulated miRNAs, such as miR-21 [74] and miR-155 [75], have emerged as dynamic biomarkers of chemoresistance: rising levels in a patient’s plasma during treatment often foreshadow therapeutic failure, prompting early transition to alternative drugs or enrollment in adaptive clinical trials. At the same time, EV-associated mRNA fragments reflecting oncogene expression and EV-lncRNAs indicating activation of resistance networks can be serially measured to guide combination strategies. For instance, detection of EV-mRNA encoding MET [76,77] or AXL [78] alongside EV-lncRNAs linked to EMT (epithelial-to-mesenchymal transition) may signal the need to add a MET inhibitor or epigenetic modulator to overcome emerging resistance. This real-time, noninvasive monitoring transforms static treatment plans into agile, biomarker-driven algorithms that continuously adapt to the tumor’s evolving molecular profile (Figure 4).

#### 2.3.3. Epigenetics

Aberrant DNA methylation patterns, particularly hypermethylation of CpG islands in tumor suppressor gene promoters, serve as powerful biomarkers for both diagnosis and therapeutic guidance [79,80]. For example, methylation of the O^6^-methylguanine-DNA methyltransferase (MGMT) promoter in glioblastoma silences DNA repair capability, rendering tumors more susceptible to alkylating agents such as temozolomide [81]. Clinicians routinely assess MGMT methylation status to identify patients most likely to derive benefit from this chemotherapy and to spare unmethylated cases from ineffective treatment. Beyond MGMT, panels of methylated genes in cell-free DNA have been developed to detect early colorectal [82] or lung cancers [83] and to stratify the risk of recurrence after surgery. Therapeutically, demethylating agents such as 5-azacytidine and decitabine reverse pathological hypermethylation, re-express silenced tumor suppressors, and sensitize leukemias and solid tumors to subsequent chemotherapy or immunotherapy [84]. Monitoring the dynamics of DNA methylation in circulating tumor DNA during treatment can therefore inform on drug activity and emerging resistance, enabling oncologists to adjust or escalate epigenetic therapy in real time, such as inter-patient variability in methylation backgrounds and the influence of inflammatory or hematopoietic cells on cfDNA methylation patterns [85].

Chromatin accessibility profiling reveals the regulatory landscapes that govern oncogene activation and drug resistance [86]. Assays such as ATAC-seq applied to tumor tissue or sorted circulating tumor cells map regions of open chromatin, pinpointing active enhancers and promoters that drive tumor proliferation or survival pathways [87]. For instance, increased accessibility at super-enhancer regions has been linked to aggressive subtypes of cancer; recognizing these accessible sites enables the development of small molecules or bromodomain inhibitors that disrupt enhancer-driven transcription [86]. Furthermore, shifts in the chromatin accessibility profile can forecast the onset of resistance: tumors adapting to hormone therapy often open alternative enhancer regions to activate bypass signaling. By serially profiling chromatin accessibility before and during treatment, researchers can identify emerging regulatory vulnerabilities and intervene with targeted epigenetic modulators, such as histone deacetylase (HDAC) [88] or bromodomain and extra-terminal motif (BET) inhibitors [89], precisely when resistance is imminent. This dynamic view of the epigenome thus offers both predictive biomarkers and direct targets for next-generation cancer therapies (Figure 4).

#### 2.3.4. Proteomics

Proteomic profiling deciphers the abundance, post-translational modifications, and interaction networks of proteins that execute oncogenic programs. However, key signaling proteins such as EGFR and AKT contain multiple phosphorylation sites, and quantifying a single phospho-epitope does not reliably indicate which pathway node should be targeted therapeutically. By applying mass spectrometry or affinity-based assays to tumor tissue, plasma, or enriched exosomes, clinicians can identify overexpressed receptor tyrosine kinases (e.g., EGFR, HER2) or activated signaling nodes (e.g., phosphorylated AKT, ERK) that constitute actionable targets [90]. For example, elevated levels of phosphorylated HER2 in breast cancer have guided the addition of HER2-directed antibody–drug conjugates [91], while quantifying PD-L1 protein on tumor and immune cells has become routine to select patients for checkpoint blockade [92]. Beyond guiding initial therapy, serial proteomic measurements during treatment can reveal adaptive rewiring of signaling pathways, prompting combination strategies that forestall resistance, such as the limited dynamic range of mass-spectrometry detection, which can hinder accurate quantification of low-abundance cytokines or phosphoproteins [93]. Emerging platforms capable of single-cell proteomics further promise to resolve intratumoral heterogeneity, ensuring that all clinically relevant protein subclones are targeted [94] (Figure 4).

#### 2.3.5. Metabolomics

Cancer cells rewire their metabolism to support rapid proliferation, survival under stress, and immune evasion [95,96]. Metabolomic analyses, using nuclear magnetic resonance (NMR), mass spectrometry, or hyperpolarized MRI, detect these alterations in tumor extracts, blood, or even urine [97]. Accumulation of oncometabolites like 2-hydroxyglutarate in isocitrate dehydrogenase (IDH)-mutant gliomas not only serves as a diagnostic biomarker but also directly informs the use of small-molecule IDH inhibitors, which can induce differentiation and clinical responses [98,99,100,101]. Elevated choline or lactate peaks on magnetic resonance (MR) spectroscopy correlate with aggressive tumor behavior and have guided dose intensification in radiotherapy [102,103]. Furthermore, longitudinal monitoring of metabolic fluxes can gauge therapeutic impact: a drop in glutamine uptake during glutaminase inhibitor therapy indicates target engagement, whereas a resurgence heralds metabolic escape, signaling the need for combination with inhibitors of compensatory pathways [104,105,106,107]. In this way, metabolomics transforms dynamic tumor metabolism into both a biomarker and a therapeutic vulnerability, such as the strong pre-analytical variability of metabolites and rapid ex vivo degradation that can mask true biological differences (Figure 4).

#### 2.3.6. Microbiomics

The composition and function of the microbiome, both in the gut and within the tumor microenvironment, profoundly influence cancer progression and treatment response [108,109]. Studies have demonstrated that the presence of specific bacterial species, such as Akkermansia muciniphila, correlates with improved outcomes on immune checkpoint inhibitors, likely by modulating systemic inflammation and antigen presentation [110]. Harnessing this insight, clinicians are exploring adjunctive strategies such as dietary interventions, probiotics, or fecal microbiota transplantation to augment immunotherapy efficacy. Within the tumor niche itself, intratumoral bacteria can metabolize chemotherapeutic agents, activating prodrugs or inactivating cytotoxics, and proteomic and genomic profiling of these microbes can predict which patients might benefit from antibiotic modulation [30,111]. As sequencing technologies mature and causal links between microbial taxa and treatment response become clearer, microbiome profiling stands to become a standard companion biomarker, guiding both direct microbial interventions and optimization of conventional therapies, such as contamination artifacts and batch effects that remain a challenge when sequencing low-biomass microbial samples (Figure 4).

## 3. Clinical Applications of Biomarkers

### 3.1. Early Detection

Detecting cancer at its earliest, most treatable stage poses a unique set of challenges: screening must be highly sensitive (minimizing false negatives even at the expense of some false positives), yet affordable and non-invasive enough to ensure broad patient adherence. Liquid biopsy, panels of methylated cell-free DNA, circulating microRNAs, or other cell-free biomarkers, offer a rapid, simple, and minimally invasive platform ideally suited to these requirements [112]. Liquid biopsy’s minimal infrastructure needs make it especially promising for low-resource settings. In addition, liquid biopsy dried blood-spot tests are under development to enable cell-free DNA profiling from a simple heel- or finger-prick sample [113], which is especially helpful in regions where even venous phlebotomy is difficult, such as rural clinics in underdeveloped countries. By combining these molecular risk scores with traditional factors (family history, histology, environmental exposures), precision oncology programs can tailor screening intervals and preventive measures to each patient’s personalized risk profile, while maintaining the affordability and accessibility necessary for global implementation (Figure 5).

### 3.2. Theranostics

Theranostic biomarkers combine diagnostic imaging with targeted therapy to deliver personalized treatment “on the same molecule.” A prime example is PSMA PET imaging in metastatic prostate cancer [114,115]. Patients whose tumors avidly uptake radiolabeled PSMA tracers can subsequently receive PSMA-targeted radionuclide therapy, thereby using the same antigen for both detection and treatment. Similarly, somatostatin receptor scintigraphy in neuroendocrine tumors identifies candidates for peptide receptor radionuclide therapy (PRRT), ensuring that only patients with sufficient receptor expression receive the therapeutic radiopeptide [116,117,118]. This biomarker-driven approach streamlines patient selection, maximizes on-target delivery, and minimizes off-target toxicity by confirming target presence before administering therapy. On the other hand, building on this foundation, emerging artificial intelligence (AI)-assisted, multi-modal phenotyping platforms integrate imaging data, genomic profiles, proteomic signatures, and clinical parameters to generate deeply personalized treatment blueprints. Machine-learning algorithms can, for instance, fuse PET/CT uptake patterns with circulating tumor DNA mutations and tumor microenvironment features to predict not only target expression but also likely resistance mechanisms, enabling dynamic adaptation of radionuclide dose or combination with immunotherapy agents [119]. Notably, this AI-driven multimodal approach mirrors the traditional Chinese medicine principle of “bian zheng shi zhi” [120], tailoring therapy to each patient’s unique molecular “syndrome” (Figure 5).

### 3.3. Monitoring and Surveillance

Once therapy is underway, dynamic biomarkers provide real-time insight into tumor burden and treatment efficacy. Effective monitoring requires multiple, longitudinal samplings, but repeated tissue biopsies carry risks: infection from the procedure [121], immunosuppression from anesthesia [122,123] (which can paradoxically promote metastasis or recurrence), and patient discomfort that undermines adherence. By contrast, non-invasive liquid biopsy sidesteps these issues. Circulating tumor DNA assays have revolutionized surveillance by allowing oncologists to track variant allele frequencies in plasma as a surrogate for tumor volume. For example, in colorectal cancer patients treated with anti-EGFR antibodies, a rise in KRAS-mutant ctDNA often precedes radiographic progression by weeks, enabling early adjustment of therapy [124,125,126]. Similarly, serial measurements of exosomal protein markers, such as elevated HER2 levels in breast cancer, can signal residual disease or early relapse, prompting timely intensification of adjuvant treatment [127,128]. For MRD (minimal residual disease) monitoring, tumor-informed assays, where bespoke panels are designed around a patient’s known genetics or epigenetic profile, offer superior sensitivity at lower cost, enabling frequent, minimally invasive checks on disease status [129]. By integrating biomarker-guided surveillance with traditional imaging schedules, clinicians can tailor follow-up intervals, minimize unnecessary scans, and intervene before overt clinical relapse (Figure 5).

### 3.4. Prion-Based Detection Modalities

Prion biology also represents an unconventional molecular detection modality. Although classically associated with neurodegenerative disorders, prion-like protein misfolding and self-propagating conformational templates can be captured through amplification assays such as RT-QuIC or PMCA, enabling ultra-sensitive readouts of pathological protein seeding activity. These assays expand the conceptual boundary of biomarker detection beyond nucleic acids and conventional proteins by measuring conformational biochemistry as a diagnostic signal.

### 3.5. Summary of Biomarker Suitability Across Clinical Contexts

Different biomarker modalities vary substantially in their appropriateness for early detection, relapse surveillance, and therapy monitoring. For early detection, assays with high sensitivity for low-burden disease, such as cfDNA methylation panels, fragmentomics, and microbiome-derived signatures, are generally most informative, whereas protein markers with low dynamic range (e.g., CEA, CA19-9) tend to perform poorly at this stage. For relapse detection, ctDNA variant tracking and structural variant detection (e.g., fusion breakpoints or CNV shifts) are particularly effective because they can reveal molecular recurrence months before radiographic progression, while metabolomics or proteomics alone are often insufficiently sensitive. For real-time therapy monitoring, dynamic biomarkers such as ctDNA allele-frequency changes, EV-RNA expression shifts, or metabolic flux alterations outperform static genomic alterations, which rarely change quickly enough to guide day-to-day treatment decisions. This framework highlights how distinct biomarker classes align with specific clinical needs along the cancer continuum.

## 4. Impact of Biomarkers on Treatment Decisions

### 4.1. Biomarker-Driven Therapeutic Target Identification

The discovery of actionable molecular alterations through biomarker analysis has transformed treatment paradigms across cancer types. For instance, sequencing of tumor DNA to detect activating mutations in kinases, such as EGFR, in lung cancer [130,131] or BRAF in melanoma [132,133], enables the direct prescription of small-molecule inhibitors that shut down the driver oncogene. In parallel, proteomic assays revealing overexpression of surface receptors, like HER2 in breast and gastric cancers, guide the use of monoclonal antibodies and antibody–drug conjugates tailored to that target [134]. Beyond single-gene alterations, multi-omics panels integrate genomic, transcriptomic, and epigenetic data to uncover less obvious vulnerabilities, such as studies suggest therapeutic targets from transcriptomic signature analysis [135]. In each case, the identification of a specific biomarker not only predicts which drug will yield the greatest benefit, but also spares patients from ineffective, broad-spectrum therapies.

### 4.2. Resistance Biomarkers

Despite initial responses, tumors frequently evolve mechanisms to evade targeted agents. Monitoring for resistance biomarkers, either through repeat tissue biopsy or more commonly via liquid biopsy, allows clinicians to stay one step ahead of disease progression. A classic example is the emergence of the EGFR T790M mutation in non-small cell lung cancer patients treated with first-generation EGFR inhibitors; its detection in circulating tumor DNA prompts a switch to third-generation inhibitors designed to overcome this specific resistance [136]. Similarly, secondary amplifications of MET or HER2 have been observed upon resistance to EGFR or ALK inhibitors, leading to combination strategies that include MET or HER2 blockade [137]. In hormone-driven cancers, mutations in the estrogen receptor gene (ESR1) flagged in plasma predict resistance to aromatase inhibitors and guide a transition to selective estrogen receptor degraders [138,139]. Biomarkers have also been linked to resistance against immunotherapies. For example, elevated expression of exhaustion markers on T cells can predict poor response to checkpoint inhibitors [36]. However, many of these candidates rest on in silico or retrospective bioinformatic analyses [140,141,142] and demand prospective clinical and functional validation before they can inform practice. By dynamically profiling resistance biomarkers, clinicians can anticipate emerging escape mechanisms and tailor sequential therapies accordingly. Such an adaptive, biomarker-driven strategy aims not only to maintain disease control but also to prolong survival by staying one step ahead of tumor evolution.

### 4.3. Early Intervention

Early screening and early detection of cancer through highly sensitive assays enable intervention at a truly preemptive stage, potentially preventing disease before it ever becomes clinically apparent [143]. For instance, methylated cell-free DNA panels can flag colorectal neoplasia at a size far below imaging resolution, allowing clinicians to initiate endoscopic or chemopreventive measures well before invasive cancer develops [144]. Similarly, monitoring for minimal residual disease (MRD) after surgery or initial therapy using ultrasensitive ctDNA assays identifies patients harboring microscopic disease that would otherwise go undetected [145]. In colorectal cancer, the presence of postoperative ctDNA signals a high risk of relapse; by stratifying these patients immediately after resection, oncologists can escalate adjuvant chemotherapy sooner than conventional clinical or radiographic criteria would allow [146,147]. By shifting from reactive treatment of overt recurrence to preemptive, marker-driven intervention, we stand to eradicate microscopic tumor foci and substantially improve long-term outcomes..

### 4.4. Real-Time Intervention upon Relapse

Once a patient enters maintenance therapy or surveillance, continuous biomarker monitoring delivers actionable insights long before clinical progression. Serial measurements of ctDNA variant allele frequencies can unmask expanding resistant clones days to weeks ahead of radiographic changes. In these cases, clinicians can immediately adapt treatment, escalating the dose, adding a second targeted agent, or switching to a different therapeutic class, to nip overt relapse in the bud. This agile, “just-in-time” strategy, unlike fixed, calendar-based schedules, keeps therapy intensity and selection tightly synchronized with the tumor’s shifting molecular profile. Because it relies on minimally invasive liquid biopsies rather than repeat tissue sampling, it minimizes procedural risk (infection, anesthesia-induced immunosuppression) and enhances patient adherence. Moreover, when paired with AI-driven predictive models that integrate longitudinal ctDNA trends, imaging data, and clinical parameters, this approach can forecast resistance trajectories and recommend optimal combination or sequencing strategies. Ultimately, real-time intervention upon molecular relapse holds the promise of maximizing efficacy, reducing unnecessary toxicity, and conserving healthcare resources by targeting treatment exactly when and where it is needed.

## 5. Translational Challenges

### 5.1. Biomarker Validation and Clinical Trial Design

Translating a promising biomarker from the laboratory bench to clinical practice requires rigorous analytical and clinical validation. Analytical validation ensures that an assay reliably measures the biomarker across different sample types, instruments, and laboratories, whereas clinical validation establishes that the biomarker predicts a meaningful clinical outcome. Designing clinical trials around biomarkers adds further complexity: enrichment strategies must identify patients who truly carry the target alteration, control arms may need to reflect standard-of-care molecular testing, and adaptive trial designs are often employed to allow midstream adjustments based on emerging biomarker data. Without careful validation and trial planning, even the most compelling molecular finding can fail to demonstrate benefit in real-world patient populations (Table 1).

### 5.2. Regulatory Approval and IVD Registration

Bringing a diagnostic biomarker into routine use typically involves approval of an in vitro diagnostic (IVD) device, which must meet stringent regulatory standards for safety and effectiveness. Developers must compile extensive technical documentation, including analytical performance, reproducibility, and clinical utility data, and navigate diverse requirements across jurisdictions. Companion diagnostics linked to specific therapeutics face even greater scrutiny, as both the drug and its biomarker assay must demonstrate a coordinated benefit–risk profile. The need to align co-development timelines for the therapeutic agent and its companion diagnostic can slow market entry, and post-market surveillance obligations add an ongoing burden for manufacturers and clinical laboratories. (Table 1)

### 5.3. Standardization and Quality Control

Even after a biomarker assay is approved, ensuring consistent performance across hundreds of clinical laboratories poses a major challenge. Variability in pre-analytical factors, such as sample collection, handling, and storage, can alter biomarker levels, while differences in assay platforms and bioinformatics pipelines may yield divergent results from the same specimen. Establishing external quality assurance programs, proficiency testing, and harmonized reference materials is critical to maintain high standards and to allow results generated at different centers to be directly comparable. In the context of multi-center trials or widespread clinical deployment, robust quality control frameworks mitigate against false positives or negatives that could lead to inappropriate treatment decisions (Table 1).

### 5.4. Privacy and Ethical Issues

Multi-omics biomarker discovery and application generate vast amounts of sensitive patient data, raising challenges in data integration, interpretation, and governance. Advanced bioinformatic methods are needed to harmonize genomic, transcriptomic, proteomic, and clinical datasets, distinguish true signals from noise, and translate complex molecular profiles into actionable reports. At the same time, safeguarding patient privacy and obtaining informed consent for diverse analyses, from tumor sequencing to microbiome profiling, requires clear ethical frameworks and compliance with data protection regulations. Equitable access to biomarker-driven therapies must also be addressed: disparities in genomic testing availability can exacerbate existing health inequities, underscoring the need for policies that ensure all patients benefit from precision oncology advances (Table 1).

### 5.5. Controversies and Consensus

Inter-laboratory discordance in variant detection, particularly between CAP- and CLIA-certified laboratories, remains a significant obstacle to harmonized patient care, as differences in DNA extraction, library prep, sequencing depth, and bioinformatic thresholds can yield divergent calls from identical cfDNA or tissue samples. To address this, cross-platform calibration using standardized reference materials (e.g., NIST’s Genome in a Bottle cell-line mixtures spiked at known allele frequencies) has been proposed to align sensitivity, specificity, and limits of detection across centers, thereby reducing false positives and negatives and enabling reliable longitudinal monitoring [148]. A parallel controversy centers on the choice of liquid versus tissue biopsy: the FDA-approved Guardant360 cfDNA assay affords non-invasive, repeatable sampling and multi-site disease surveillance but may lack sensitivity in low-shedding lesions and cannot pinpoint mutation origin, whereas tissue-based NGS delivers histopathological context and structural variant detection at the cost of sampling bias and procedural risk. In recognition of these complementary strengths and weaknesses, the 2023 ESMO guidelines recommend a dual-validation strategy (initial broad-spectrum cfDNA screening followed by confirmatory tissue NGS for actionable variants and precise lesion localization) to maximize diagnostic yield while mitigating each modality’s limitations [149] (Table 1).

### 5.6. Leakage and the Reproducibility Crisis in Machine-Learning-Based Biomarkers

Machine learning holds great promise for uncovering complex, high-dimensional biomarker signatures, but its apparent breakthroughs are often undermined by a recurring problem: data leakage [150]. When information from hold-out samples, whether through premature feature selection, inadvertent use of future timepoints, or tuning hyperparameters on test-set performance, seeps into model training, reported accuracies become dangerously overoptimistic and fail to replicate in independent cohorts. To guard against these pitfalls, investigators have proposed the use of “Model Information Sheets” [150]: standardized templates that require investigators to detail every step of their workflow, from how samples are split and preprocessed to when and where features are chosen and models are calibrated. Although most demonstrations of leakage arise from methodological research rather than prospective biomarker trials, parallel analyses in other domains, such as civil-war prediction, where correcting for leakage erased the alleged superiority of complex ML architectures, underscore the real-world stakes of lax validation. Embedding similar reproducibility checks into biomarker studies, especially those destined for clinical translation, will be essential to ensure that machine-learning innovations truly deliver on their promise rather than simply reflecting self-fulfilling analytical artifacts (Table 1).

## 6. Personalized Precision Oncology Strategies

### 6.1. Intra- and Inter-Tumor Heterogeneity

Tumors are not monolithic entities but rather ecosystems of diverse cell populations harboring distinct molecular alterations. This heterogeneity exists both within a single lesion (intra-tumor) and between primary and metastatic sites (inter-tumor), driving variable therapeutic responses and the emergence of resistant clones. For example, a basal-like subclone in a predominantly luminal breast cancer may lack HER2 amplification, rendering anti-HER2 therapy less effective against that compartment [151]. Recognizing this diversity, clinicians are now employing multi-region biopsies and longitudinal liquid biopsies to capture the full landscape of tumor variants. By mapping the clonal architecture, treatment regimens can be designed to target the most aggressive or therapy-resistant subpopulations, often through combination therapies that simultaneously inhibit multiple pathways.

### 6.2. Depth and Breadth of Detection: Multi-Omics Integration

Effective personalization demands both “depth”, comprehensive characterization of a tumor’s key drivers, and “breadth”, assessment across multiple molecular layers. Integrating genomic, transcriptomic, epigenomic, proteomic, and metabolomic data yields a holistic view of tumor biology, revealing vulnerabilities that may be missed by single-modality assays. In practice, a patient’s tumor may be profiled with whole-exome sequencing to identify driver mutations, RNA-seq to quantify fusion transcripts, ATAC-seq to map open chromatin regions, and mass spectrometry to detect overactive signaling proteins. The convergence of these datasets through advanced bioinformatics then guides a multi-agent treatment plan. Such multi-omics strategies ensure that precision oncology addresses the complexity of each individual’s disease.

### 6.3. Cost-Effectiveness and Clinical Accessibility

Despite the promise of comprehensive molecular profiling, cost and resource constraints often limit access to precision medicine. Extensive multi-omics assays can be prohibitively expensive and require specialized infrastructure, creating disparities between well-resourced centers and community clinics. To balance depth of information with economic feasibility, tiered testing algorithms have emerged: initial screening with targeted mutation panels identifies the majority of actionable alterations, while reflex testing with broader assays is reserved for patients without clear targets. Pooled testing approaches, such as multiplexed liquid biopsy panels, can further reduce per-patient costs. Health-economic studies are increasingly essential to demonstrate that upfront investment in biomarker testing leads to downstream savings by avoiding ineffective treatments, reducing hospitalizations, and prolonging survival, thereby supporting reimbursement and wider implementation.

### 6.4. AI and Big-Data-Driven Personalization

The explosion of high-dimensional molecular and clinical data has created fertile ground for artificial intelligence (AI) and machine-learning applications in oncology. Predictive models trained on large datasets can identify subtle biomarker patterns that correlate with drug response or toxicity, enabling treatment recommendations that evolve as new real-world evidence accumulates. AI-driven platforms also facilitate real-time clinical decision support: integrating electronic health records, imaging features, and multi-omics profiles to generate personalized therapeutic options at the point of care. Moreover, federated learning approaches promise to overcome data privacy barriers, allowing institutions to collaboratively refine predictive algorithms without sharing raw patient data. As these technologies mature, they will transform precision oncology from a static test-and-treat model into a dynamic, continuously learning system that adapts to each patient’s unique disease trajectory.

## 7. Conclusions

This review has highlighted the spectrum of cancer biomarkers, from tissue-based assays and liquid biopsy modalities to multi-omics and imaging markers, and their pivotal roles in guiding precision oncology. We have seen how DNA alterations, RNA signatures, epigenetic modifications, proteomic profiles, metabolic changes, and even microbiome composition inform treatment selection, monitoring, and adaptation. Biomarker-driven target identification has enabled the deployment of highly effective therapies such as kinase inhibitors, monoclonal antibodies, and immune modulators, while dynamic monitoring of resistance markers and minimal residual disease has ushered in an era of truly adaptive treatment strategies. Moreover, the integration of multi-dimensional data has revealed complex tumor heterogeneity and uncovered novel vulnerabilities, underscoring the importance of both depth and breadth in molecular profiling.

Despite these advances, significant hurdles remain before the full promise of biomarker-guided oncology can be realized. Many candidate biomarkers lack robust analytical and clinical validation, and the high cost and logistical complexity of multi-omics assays limit their widespread adoption. Regulatory pathways for companion diagnostics and standardization across laboratories continue to lag behind scientific innovation, creating bottlenecks in clinical deployment. Additionally, the dynamic interplay between tumor cells and the host microenvironment, including immune components and the microbiome, remains incompletely understood, leaving gaps in our ability to predict and overcome therapy resistance. Finally, ethical, privacy, and equity considerations surrounding the collection and use of high-dimensional patient data demand ongoing attention to ensure responsible and inclusive precision medicine.

Looking ahead, advancing precision oncology will require concerted efforts to streamline biomarker validation through adaptive trial designs and real-world evidence generation. The development of cost-effective, scalable platforms, such as integrated liquid biopsy panels and standardized bioinformatics pipelines, will be critical to democratize access. Artificial intelligence and federated learning hold promise to harness disparate, multi-institutional datasets while preserving patient privacy, enabling more accurate predictive models and decision-support tools. On the therapeutic front, combinations of targeted agents, epigenetic modulators, and immunotherapies guided by real-time biomarker feedback will likely become the norm. Finally, closer integration of patient-derived organoids, single-cell analyses, and spatial profiling techniques will deepen our understanding of tumor ecology and accelerate the translation of novel biomarkers into actionable clinical tests. Through these innovations, the vision of truly personalized, adaptive cancer care will continue to move from possibility to reality.

## Figures and Tables

**Figure 1 cancers-17-03720-f001:**
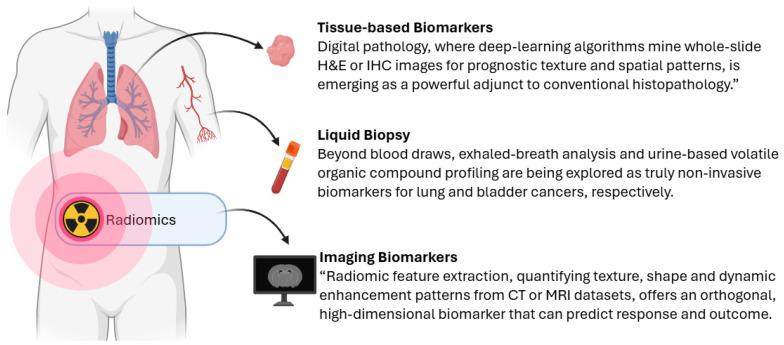
Sample types and their associated biomarker modalities. Schematic representation of three primary cancer sample sources (tumor tissue, blood, and medical imaging) and their corresponding biomarker readouts. Several biomarkers have already been validated in this cancer type and sample source. As described in the main text, several clinically validated biomarkers exemplify these detection windows. For instance, TERT promoter mutations and TP53 R249S in circulating cell-free DNA, cfDNA-based assays, as well as serum markers such as AFP, AFP-L3, and DCP for hepatocellular carcinoma. These examples (with supporting references provided in the text) illustrate how the framework depicted here corresponds to real-world biomarker applications.

**Figure 2 cancers-17-03720-f002:**
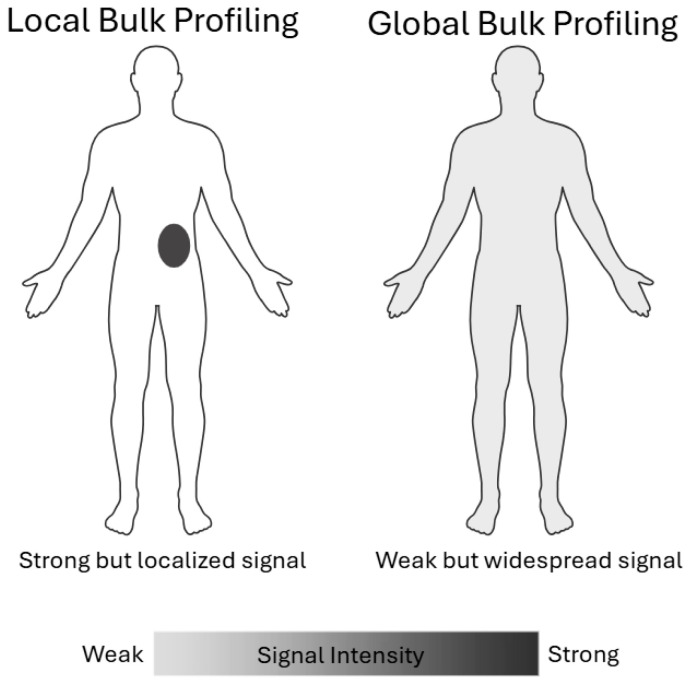
Conceptual comparison between local and global bulk profiling strategies. Local bulk profiling captures strong molecular signals from a specific anatomical region or lesion, offering high signal intensity but limited spatial scope. In contrast, global bulk profiling integrates signals across broader or systemic regions, resulting in more widespread but diluted signals. This trade-off between intensity and spatial resolution has important implications for biomarker sensitivity and disease monitoring, especially in heterogeneous conditions such as cancer or inflammation.

**Figure 3 cancers-17-03720-f003:**
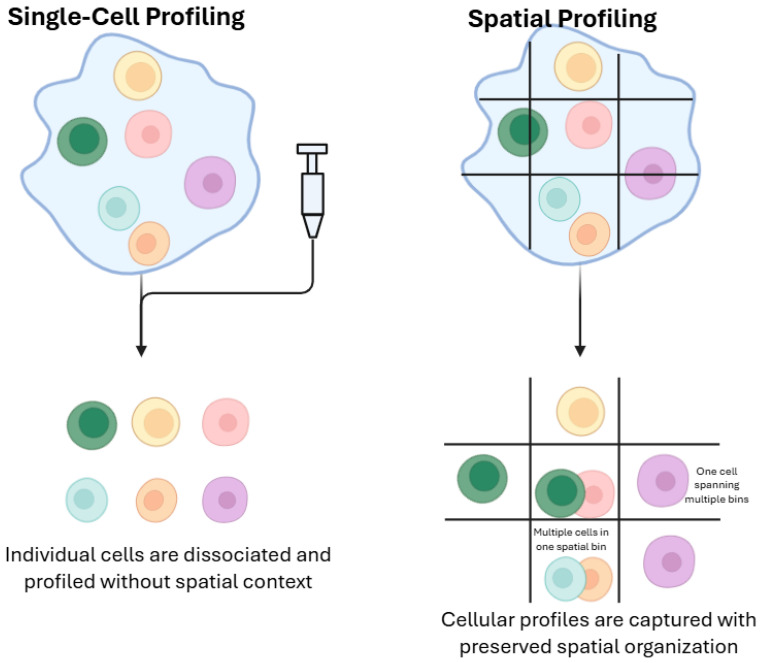
Comparison between single-cell and spatial profiling approaches. Single-cell profiling (**left**) involves dissociating individual cells from tissue, enabling high-resolution molecular characterization but resulting in the loss of spatial context. In contrast, spatial profiling (**right**) retains the physical organization of cells within the tissue architecture, allowing molecular features to be linked to specific spatial locations. While single-cell approaches provide detailed cell-level information, spatial methods enable mapping of cell states, interactions, and microenvironments within their native context.

**Figure 4 cancers-17-03720-f004:**
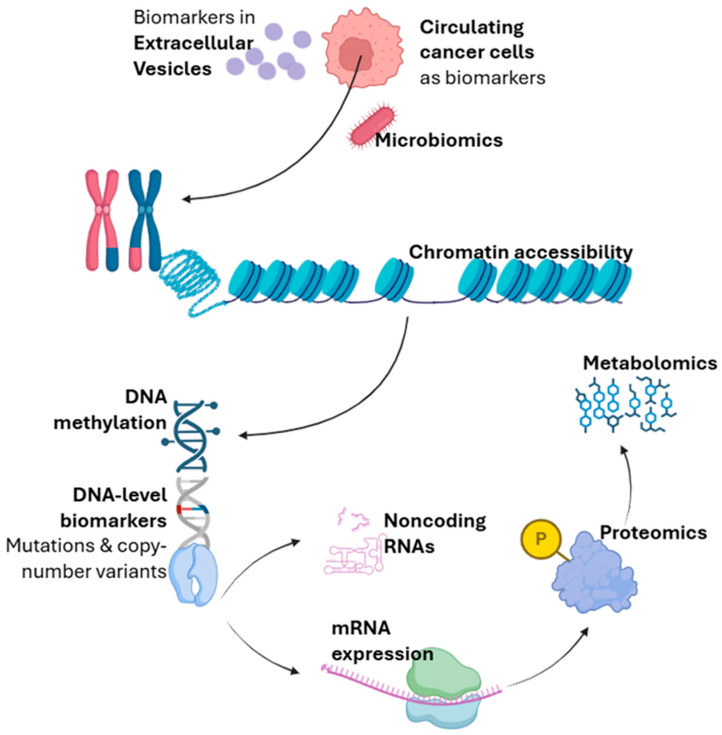
Schematic of molecular detection modalities for cancer biomarkers. Tumor-derived materials are sampled and profiled across multiple molecular layers: Extracellular Vesicles & Circulating Cancer Cells (**top**): Vesicle-encapsulated DNA, RNA, and protein cargo, alongside intact tumor cells, enable noninvasive capture of structural variants, transcriptomes, and proteomes. Microbiomics (**top right**): Commensal and intratumoral microbial DNA and metabolites inform host–microbe interactions that modulate treatment response. DNA-Level Biomarkers (**middle left**): Point mutations, copy-number variants, and DNA methylation patterns are detected in tissue or cell-free DNA. Chromatin Accessibility (**middle right**): Open chromatin regions mapped by ATAC-seq or similar assays reveal active regulatory elements driving oncogene expression. Transcriptomics (**bottom left**): mRNA abundance and noncoding RNA (miRNA, lncRNA) profiles reflect real-time oncogenic and resistance programs. Proteomics & Metabolomics (**bottom right**): Mass spectrometry-based quantification of proteins (including post-translational modifications) and small-molecule metabolites captures functional and metabolic states of the tumor.

**Figure 5 cancers-17-03720-f005:**
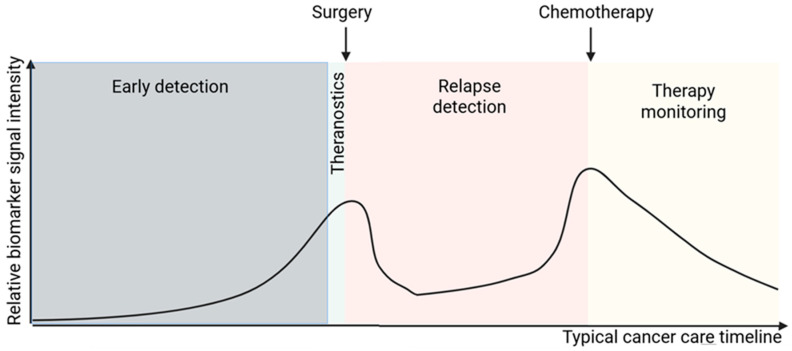
Applications of biomarkers across a typical cancer therapeutic process. The schematic plots relative biomarker signal intensity over key phases of a typical cancer therapeutic pathway (x-axis). Early detection (gray): screening and diagnosis via high-sensitivity assays. Theranostics (teal): target identification immediately before surgery (arrow) to guide both imaging and radionuclide therapy. Relapse detection (pink): post-surgical surveillance for minimal residual disease. Therapy monitoring (yellow): real-time assessment during chemotherapy (arrow) to adjust treatment.

**Table 1 cancers-17-03720-t001:** Key Translational Challenges in Biomarker Development and Clinical Implementation.

Aspect	Key Issue	Major Challenges	Current Mitigation Strategies
Biomarker Validation and Clinical Trial Design	Moving from discovery to clinical utility via analytical and clinical validation; biomarker-driven trial design (e.g., enrichment, adaptive trials)	Misclassification of biomarker-positive patients; complexity of trial design; limited generalizability	Enrichment and adaptive trial designs; clearly defined predictive value; appropriate control arms
Regulatory Approval and IVD Registration	In vitro diagnostic (IVD) registration requires demonstration of safety and effectiveness; companion diagnostics must co-develop with therapeutics	Regulatory heterogeneity across regions; burden of technical documentation; timeline mismatch with drug development	Early coordination of drug-CDx timelines; compliance with FDA/EMA/CE marking standards
Standardization and Quality Control	Ensuring reproducibility across sites; minimizing pre-analytical and analytical variability	Inter-lab variability in results; lack of harmonized materials and protocols	External quality assurance, proficiency testing, use of reference materials (e.g., NIST standards)
Privacy and Ethical Issues	Handling large-scale multi-omics data; protecting patient privacy and consent; ensuring equity	Data integration complexity; regulatory requirements for consent and access; disparities in test availability	Ethical frameworks for consent and data sharing; compliance with GDPR/HIPAA; policies to reduce access gaps
Controversies and Consensus	Discordance between cfDNA and tissue-based assays; inter-lab variant detection inconsistency	False positives/negatives due to methodological differences; biopsy type trade-offs	Standardization using spiked reference materials; dual-validation approach (cfDNA screening + tissue confirmation) per 2023 ESMO guidelines
Leakage and Reproducibility Crisis in ML-Based Biomarkers	Overfitting and data leakage in ML biomarker models lead to inflated performance	Use of test data in model training; lack of transparency in workflow reporting	Adoption of Model Information Sheets; workflow auditing; emphasis on independent validation and reproducibility checks

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
