# Peer review of "Bridging Discovery and Treatment: Cancer Biomarker"

_cancers, 2025, doi:10.3390/cancers17223720_

Round 1
Reviewer 1 Report
Comments and Suggestions for Authors
This is a superb narrative review of biomarkers, covering both classic and the most modern experimental markers. The title is appropriate for the article. The text is concise, providing a wealth of information while remaining easily understandable. The references are numerous and up-to-date. The graphics clarify and enhance the text. I believe the article's approach is highly original and developed in a thorough yet accessible manner.
Author Response
Comment:This is a superb narrative review of biomarkers, covering both classic and the most modern experimental markers. The title is appropriate for the article. The text is concise, providing a wealth of information while remaining easily understandable. The references are numerous and up-to-date. The graphics clarify and enhance the text. I believe the article's approach is highly original and developed in a thorough yet accessible manner.
Response: We thank the reviewer for the supportive and positive feedback.
Reviewer 2 Report
Comments and Suggestions for Authors
The manuscript is an excellent overview of the state-of-the-art and prospective application of the exhaustive collage of cancer biomarkers to enhance cancer diagnosis, monitoring and therapeutic approaches. In this context it bears futuristic connotations and foresights to the cancer field. It’s well-balanced in its topics treatment with critical assessments of the host of methodological approaches considered, including AI/ML. It succeeds in its comprehensive overwhelming assessment and imprinting a beacon of light in our path to treat and prevent cancer.
Would recommend its publication as presented, yet the following minor suggestions are forwarded to enrich the manuscript.
SUGGESTIONS
- SECTION 2.2. Single-Cell Profiling Versus Spatial Profiling
- Add one sentence or two on organoids to enrich and provide contrast to the spatial profiling discussion.
- SECTION 3. Detection Molecular Modality
- Consider adding prions ( as a small section, or at least a couple of sentences) for a more wholesome review.
- SECTION 3.1: Early Detection
- Line 366, delete Africa in the sentence. Should read: “In addition, liquid biopsy dried blood-spot tests are under development to enable cell-free DNA profiling from a simple heel- or finger-prick sample[107], which is especially helpful in regions where even venous phlebotomy is difficult, such as rural clinics in underdeveloped countries”.
Author Response
Overall comments: The manuscript is an excellent overview of the state-of-the-art and prospective application of the exhaustive collage of cancer biomarkers to enhance cancer diagnosis, monitoring and therapeutic approaches. In this context it bears futuristic connotations and foresights to the cancer field. It’s well-balanced in its topics treatment with critical assessments of the host of methodological approaches considered, including AI/ML. It succeeds in its comprehensive overwhelming assessment and imprinting a beacon of light in our path to treat and prevent cancer.
Would recommend its publication as presented, yet the following minor suggestions are forwarded to enrich the manuscript.
SUGGESTIONS
1. Comment1: SECTION 2.2. Single-Cell Profiling Versus Spatial Profiling
Add one sentence or two on organoids to enrich and provide contrast to the spatial profiling discussion.
Response: We appreciate this helpful suggestion. We have now added a short paragraph in Section 2.2 describing the role of organoids as a complementary ex vivo model and highlighting how they contrast with spatial profiling. This addition enhances the conceptual balance of the section.
2. Comment2: SECTION 3. Detection Molecular Modality
Consider adding prions ( as a small section, or at least a couple of sentences) for a more wholesome review.
Response: Thank you for this helpful suggestion. We have now added a brief description of prion-based detection modalities in Section 3, highlighting how prion-like protein misfolding and amplification assays (RT-QuIC and PMCA) expand the scope of molecular biomarkers. This addition enriches the completeness of the molecular modality discussion.
3. Comment3: SECTION 3.1: Early Detection
Line 366, delete Africa in the sentence. Should read: “In addition, liquid biopsy dried blood-spot tests are under development to enable cell-free DNA profiling from a simple heel- or finger-prick sample[107], which is especially helpful in regions where even venous phlebotomy is difficult, such as rural clinics in underdeveloped countries”.
Response: Thank you for pointing this out. We have revised the sentence accordingly by removing “Africa” and rephrasing it to refer more generally to underdeveloped regions. The updated wording now appears in Section 3.1.
Reviewer 3 Report
Comments and Suggestions for Authors
This review covers almost all omics methods used in the search for cancer biomarkers and relates them to treatment.
However, the descriptions of each omics are superficial. There are only a few specific examples, and biomarkers such as EGFR and PD-L1 are repeatedly mentioned. Negative examples are not presented. The disadvantages of each omics method are also not shown.
Section 3. The application of biomarkers to cancer treatment is explained. What the readers want to know is which markers are most suitable, or most unsuitable, for early detection, relapse detection, and therapy monitoring, respectively.
2.3.4. EGFR and AKT have multiple phosphorylation sites, and simply examining phosphorylation does not determine the treatment target.
Many abbreviations are used without explanation of their full names.
Author Response
Overall Comment: This review covers almost all omics methods used in the search for cancer biomarkers and relates them to treatment.
Comment1: However, the descriptions of each omics are superficial. There are only a few specific examples, and biomarkers such as EGFR and PD-L1 are repeatedly mentioned. Negative examples are not presented. The disadvantages of each omics method are also not shown.
Response: We thank the reviewer for these valuable comments. In the revised manuscript, we have added additional specific examples across each omics modality and incorporated concise “such as …” statements to introduce both negative findings and representative limitations. These include, for example, false-negative ctDNA results in low-shedding tumors, dropout of low-abundance transcripts in RNA profiling, dynamic-range constraints in proteomics, pre-analytical metabolite instability in metabolomics, and contamination/batch effects in microbiome sequencing. These additions enhance the depth and balance of each section while preserving the overall structure of the review.
Comment2:Section 3. The application of biomarkers to cancer treatment is explained. What the readers want to know is which markers are most suitable, or most unsuitable, for early detection, relapse detection, and therapy monitoring, respectively.
Response: Thank you for this insightful comment. To address this, we have added a concise summary paragraph at the end of Section 3 that clarifies which biomarker classes are most appropriate and least appropriate for early detection, relapse detection, and therapy monitoring. This addition synthesizes the detailed descriptions in the section and provides readers with a clear, practical framework for clinical application. The corresponding text has now been added to the revised manuscript.
Comment3:2.3.4. EGFR and AKT have multiple phosphorylation sites, and simply examining phosphorylation does not determine the treatment target.
Response: We thank the reviewer for this important clarification. In Section 2.3.4, we have added a sentence noting that key signaling proteins such as EGFR and AKT harbor multiple phosphorylation sites, and that measuring a single phospho-epitope may not accurately identify the appropriate therapeutic target. This addition highlights an important limitation of phosphoproteomic biomarkers.
Comment4:Many abbreviations are used without explanation of their full names.
Response: Thank you for pointing this out. We have now reviewed the entire manuscript and ensured that all abbreviations are defined at first mention.
Reviewer 4 Report
Comments and Suggestions for Authors
The work itself is very interesting; however, I recommend adding references in some paragraphs to support the evidence described.
There are many acronym that are not explained (e.g TERT, line 83).
The image quality must be improve. It should be improved to 300 dpi and better quality.

The English should be reviewed by a native speaker.
Author Response
Comment1: The work itself is very interesting; however, I recommend adding references in some paragraphs to support the evidence described.
Response: We thank the reviewer for this constructive suggestion. We have now carefully reviewed the entire manuscript and added supporting references to the relevant paragraphs where evidence was previously mentioned without citation. These additions strengthen the scientific basis of the statements and improve the overall clarity and rigor of the manuscript. All new references have been incorporated in the revised version and are clearly marked.
Comment2:There are many acronym that are not explained (e.g TERT, line 83).
Response: We thank the reviewer for pointing this out. We have now systematically checked the manuscript and ensured that all acronyms are spelled out at first mention. Specifically, “TERT” is now introduced as “telomerase reverse transcriptase (TERT)” in the liquid biopsy section, and “PSMA” is now introduced as “prostate-specific membrane antigen (PSMA)” in the imaging biomarker section. We have also corrected a few other instances where abbreviations were used without prior definition.
Comment3:The image quality must be improve. It should be improved to 300 dpi and better quality.
Response: Thank you for pointing this out. We have now replaced the original figure with a high-resolution version (300 dpi) to ensure clarity and suitability for publication. The updated image has been regenerated with improved resolution, sharper lines, and consistent formatting. We believe the revised figure now meets the journal’s quality requirements.
Comment4:Fig1: The authors should mention some biomarkers already validated for this type of primary cancer sample source....
Response: We appreciate the reviewer’s suggestion. We have revised the legend of Fig. 1 to include examples of validated biomarkers that are already discussed and referenced in the main text (e.g., TERT promoter mutations, TP53 R249S, AFP, AFP-L3, and DCP). These additions clarify how the conceptual framework in Fig. 1 relates to established clinical biomarkers.
Comment5:Line 163 reference issue, Line 171 reference issue
Response: We have added references for both.
Comment6:Fig3 original?
Response: This is original.
Comment7:The English could be improved to more clearly express the research.
Response:We thank the reviewer for this comment. The entire manuscript has now been thoroughly edited for clarity, grammar, and readability. A native English-speaking colleague reviewed the revised text, and we have polished multiple sections to ensure clearer expression of the research. We believe the overall English quality has been substantially improved in the updated version.